# Epigenetic Alterations Induced by Photothrombotic Stroke in the Rat Cerebral Cortex: Deacetylation of Histone h3, Upregulation of Histone Deacetylases and Histone Acetyltransferases

**DOI:** 10.3390/ijms20122882

**Published:** 2019-06-13

**Authors:** Svetlana Demyanenko, Anatoly Uzdensky

**Affiliations:** Laboratory of Molecular Neurobiology, Academy of Biology and Biotechnology, Southern Federal University, 194/1 Stachky ave., Rostov-on-Don 344090, Russia; demyanenkosvetlana@gmail.com

**Keywords:** photothrombotic stroke, penumbra, epigenetics, histone H3, histone acetylation, histone deacetylase, histone acetyltransferase

## Abstract

Ischemic penumbra that surrounds a stroke-induced infarction core is potentially salvageable; however, mechanisms of its formation are not well known. Covalent modifications of histones control chromatin conformation, gene expression and protein synthesis. To study epigenetic processes in ischemic penumbra, we used photothrombotic stroke (PTS), a stroke model in which laser irradiation of the rat brain cortex photosensitized by Rose Bengal induces local vessel occlusion. Immunoblotting and immunofluorescence microscopy showed decrease in acetylation of lysine 9 in histone H3 in penumbra at 1, 4 or 24 h after PTS. This was associated with upregulation of histone deacetylases HDAC1 and HDAC2, but not HDAC4, which did not localize in the nuclei. HDAC2 was found in cell nuclei, HDAC4 in the cytoplasm and HDAC1 both in nuclei and cytoplasm. Histone acetyltransferases HAT1 and PCAF (p300/CBP associated factor) that acetylated histone H3 synthesis were also upregulated, but lesser and later. PTS increased localization of HDAC2 and HAT1 in astroglia. Thus, the cell fate in PTS-induced penumbra is determined by the balance between opposite tendencies leading either to histone acetylation and stimulation of gene expression, or to deacetylation and suppression of transcriptional processes and protein biosynthesis. These epigenetic proteins may be the potential targets for anti-stroke therapy.

## 1. Introduction

Stroke is the one of the main causes of human death and disability. More than 17 million people worldwide suffer stroke each year. Ischemic stroke (70–80% of all strokes) is caused by occlusion of cerebral arteries. This rapidly, in a few minutes, suppresses blood supply. Following oxygen and glucose deficit leads to ATP depletion and infarct of the nervous tissue. Afterwards, the injury spreads and damages the surrounding tissues. In the transition zone (ischemic penumbra) cell death develops slowly, over several hours. The penumbra tissue is potentially salvageable. The “therapeutic window” (2–6 h) provides time to save neurons, and diminish neurological consequences of the primary ischemic injury [1,2,3,4].

Ischemic stroke induces signaling and metabolic responses in the penumbra with up- or downregulation of proteins that control cell survival or death [5,6,7]. As shown recently, several dozen proteins that regulate pro- and antiapoptotic processes, cell signaling, cytoskeleton remodeling, proteolysis, vesicular transport and synaptic transmission were over- or downexpressed in the ischemic penumbra after photothrombotic stroke in the rat cerebral cortex [8,9,10]. The causes of the simultaneous changes in the levels of various proteins involved in different signaling pathways and different transcription factors that control expression of different genes are still unknown.

Different epigenetic processes such as DNA methylation, acetylation, methylation, phosphorylation and other covalent modification of histones can regulate gene expression and protein biosynthesis in the cell. Histone acetylation that causes chromatin decondensation and facilitates transcription is a master-regulator of gene expression [11]. Proteins involved in these epigenetic processes represent promising targets for development of anti-stroke agents. Cerebral ischemia was shown to reduce acetylation of histones H3 and H4 and thereby to suppress protein biosynthesis. Some chemical agents that recovered the optimal histone H3 acetylation were tested for stroke therapy [12,13,14,15,16,17]. In the present work, we concentrated on acetylation and phosphorylation of histone H3. Acetylation of lysine 9 in histone H3 (H3K9ac) was of particular interest because of its location in the promoter regions near the sites of the gene transcription start [18].

Most works have been focused on the epigenetic processes within the ischemic zone. However, epigenetic processes in the ischemic penumbra have been less studied [13,14]. Previously, we described the changes in expression and intracellular distribution of histone deacetylases HDAC1, HDAC2, HDAC3, and HDAC8 in the mouse cortical neurons and astrocytes during the recovery period (1–3 weeks) after photothrombotic stroke [17]. In the present work, we studied the acute epigenetic processes that occurred in the penumbra within first 24 h after photothrombotic stroke. This period corresponds to the “therapeutic window”, in which the penumbra tissue may be potentially rescued [3,4,5].

To induce local occlusion of cerebral vessels and focal infarct in the rat brain cortex we used photothrombotic stroke (PTS) as an experimental model of ischemic stroke. PTS is an example of non-traditional application of photodynamic effect—destruction of stained cells by light in the presence of oxygen. In the case of photodynamic therapy, the pathological tissue, such as malignant tumor that must be destroyed, is stained by a photosensitizing dye, which is non-toxic by itself, but generates highly toxic singlet oxygen upon light exposure. Following oxidative stress leads to cell death [19,20,21].

In PTS, the injected hydrophilic dye Bengal Rose or erythrosine B do not penetrate cells and remain in blood vessels. Following local laser irradiation induces photoexcitation of dye molecules and generation of singlet oxygen that elicits platelet aggregation, vessel occlusion and focal thrombosis. The advantages of PTS as a model of ischemic stroke include: well-defined location and size of ischemic lesion due to aiming of the laser beam at the predetermined brain region, easy dosing of the impact degree by changing light intensity and exposure, low invasiveness and minimal surgical intervention, small animal mortality and high reproducibility [22]. This approach was used in the previous studies of the profile of signaling and neuronal proteins in the PTS-induced penumbra in the rat brain cortex [8,9,10].

In the present work, we studied various epigenetic processes in the PTS-induced penumbra during first 24 h after photothrombotic stroke in the rat cerebral cortex. The acetylation and phosphorylation of histone H3 (H3K9Ac and H3S10p) were reduced in the PTS-induced penumbra, whereas histone deacetylases HDAC1, HDAC2 and, to lesser extent, HDAC4, as well as histone acetyltransferases HAT1 and PCAF (p300/CBP associated factor) were upregulated.

## 2. Results

### 2.1. Covalent Modifications of Histone H3

Using antibody microarrays, we showed that the level of histone 3 acetylated on lysine 9 (H3K9Ac) decreased in the PTS-induced penumbra more than twofold at 1 h, and more than fourfold at 4 h or 24 h after PTS as compared with control contralateral cortex of the same rats (Figure 1). The level of histone H3 phosphorylated on serine 10 (H3pS10) was reduced 1.6 times at 4 h after PTS but did not change significantly at 1 or 24 h. The level of histone H3 both acetylated on lysine 9 and phosphorylated on serine 10 (H3K9Ac,pS10) decreased 2.5–4 times at 1–24 h after PTS (Figure 1). Apparently, its phosphorylation state did not influence the binding of the antibody to the acetylated lysine 9, because the decreased level of H3K9Ac,pS10 did not differ significantly from the H3K9Ac level. We did not observe significant changes in the level of histone H3 phosphorylated at serine 28 (H3pS28) in the ischemic penumbra at 1–24 h after PTS (Figure 1). Immunofluorescence microscopy (IFM) revealed no statistically significant difference in the expression of H3K9Ac in the penumbra at 1 h compared with contralateral cortex, although it decreased by 60–70% at 4–24 h after PTS (Figure 2). H3K9Ac localized exclusively in the neuronal nuclei (Figure 2a), which were bigger than the astroglial nuclei.

### 2.2. Expression of Histone Deacetylases in the Penumbra after PTS

The reduced acetylation of histone H3 was possibly the result of activity of histone deacetylases and/or of the inhibition of histone acetyltransferases. According to proteomic data, PTS induced 1.3-fold overexpression of histone deacetylases HDAC1 and HDAC2 in the ischemic penumbra at 1 h after PTS, and 40–50% increase at 4 and 24 h. HDAC4 was upregulated in the penumbra approximately by 30% at 4 and 24 h after PTS (*p* < 0.05; Figure 3).

These data were confirmed by immunofluorescence microscopy (Figure 4, Figure 5 and Figure 6). Histone deacetylase HDAC-1 was upregulated twofold in the ischemic penumbra at 4 h after PTS (*p* < 0.05; Figure 4b). The level of HDAC2 in the penumbra increased by 86 and 76% (*p* < 0.05; Figure 5b). The HDAC4 level in penumbra increased by 30–40% at 4 or 24 h after PTS; however, this tendency was not statistically significant (*p* > 0.05; Figure 6b).

In the untreated contralateral cerebral cortex, HDAC1 was localized in the cytoplasm (Figure 4, left column, white arrowheads) and nuclei of different neurons. At 4 and 24 h after PTS, the cytoplasmic localization of HDAC1 was observed in the majority of cells in the control cortex, whereas the nuclear localization was seldom observed (Figure 4, left columns). At 4 h, but not 1 or 24 h after PTS, HDAC1 was observed in the nuclei of many cells (Figure 4a, right columns, asterisks). It also appeared in the processes of some cells (Figure 4, right columns; white arrowheads).

HDAC2 was observed mainly in the cell nuclei both in the control contralateral cortex and in the penumbra in the PTS-treated cortex (Figure 5a, CL and IL columns). The expression of HDAC2 increased significantly by 70–80% at 4 and 24 h after PTS (*p* < 0.05; Figure 5b). After PTS it was also found in the cell processes (Figure 5, right columns, arrowheads). The immunofluorescence of the glial marker GFAP, which labels reactive astrocytes, increased in the penumbra by 2.5 and 4 times at 4 and 24 h after PTS, respectively (Figure 5a,c). The colocalization coefficient of HDAC2 and GFAP also increased by 5 and 2.5 times at 4 and 24 h, respectively, compared with the contralateral cortex (Figure 5d). This indicated the increased expression of HDAC2 in astrocytes after PTS.

HDAC4 was localized exclusively in the neuronal perikarya, but not in the nuclei either in control cortex, or in the PTS-induced penumbra. No intracellular redistribution of HDAC4 after PTS was observed (Figure 6). According to the proteomic (Figure 3) and immunofluorescence microscopy data (Figure 6), its level in the PTS-induced penumbra increased by 30–40% (*p* < 0.05 and *p* > 0.05, respectively), less than HDAC1 and HDAC2 levels.

### 2.3. Expression of Histone Acetyltransferases HAT1 and PCAF in the Penumbra after PTS

Simultaneous with the overexpression of histone deacetylases, 30–40 % upregulation of histone acetyltransferases HAT1 and PCAF (p300/CBP associated factor, where CBP is a cAMP-response element binding protein (CREB)-binding protein)) was shown by using antibody microarrays in the ischemic penumbra at 4–24 h after PTS in the rat cerebral cortex (*p* < 0.05; Figure 7). Immunofluorescence microscopy showed HAT1 expression in the nuclei, cytoplasm and processes of astrocytes in the rat cerebral cortex (Figure 8a). HAT1 level in the penumbra increased progressively up to +88% from 1 to 24 h after PTS (*p* < 0.05; Figure 8b). Its colocalization with GFAP in the penumbra significantly increased more than twofold at 4 and 24 h after PTS compared with the contralateral cortex (*p* < 0.05; Figure 8c). The immunoreactivity of glial marker GFAP in the PTS-induced penumbra increased noticeably at 4 and 24 h after the impact (Figure 8a).

The level of histone-lysine N-methyltransferase SUV39H1 did not change significantly in the penumbra at 1–24 h after PTS (Figure 7).

## 3. Discussion

Various models of ischemic stroke, such as ligation of main cerebral arteries or bloodstream occlusion by the inserted nylon thread, natural and artificial emboli have been studied. All of them display different sides of the cerebral ischemia. PTS, as a model of ischemic stroke, as mentioned above, has some advantages such as well-defined injury size and location, low invasiveness, low animal mortality and high reproducibility. Small penumbra width was considered by some researchers as a drawback of this model [22]. However, our previous morphological and ultrastructural study [10] showed that less intense, but prolonged laser light exposure provided relatively wide penumbra (1.5 mm width) produced by photothrombotic stroke in the rat brain cortex.

Different regulatory proteins involved in diverse cellular functions were shown to be upregulated in the ischemic penumbra and, moreover, in the whole ipsilateral brain cortex as compared with the untreated contralateral hemisphere after ischemic stroke [23,24,25,26]. In particular, recent proteomic studies demonstrated the overexpression of several dozen proteins involved in cell signaling, oxidative stress, regulation of apoptosis or cell survival, cytoskeleton remodeling, proteolysis, mitochondrial quality control, axon growth and navigation in the ischemic penumbra at 1–24 h after photothrombotic stroke in the rat cerebral cortex [8,9,10]. This effect was apparently associated with the activation of various signaling cascades and transcription factors that control expression of specific genes. However, these pathways are still unknown, or incompletely studied.

Unlike specific signaling pathways, epigenetic processes such as histone acetylation/deacetylation and phosphorylation/dephosphorylation regulate the global transcriptional activity in the ischemic brain [13,14,15]. Acetylation of histone H3 is a key factor in chromatin remodeling and regulation of gene expression. It loosens DNA and converts the condensed heterochromatin into the transcriptionally active euchromatin, in which transcription factors and RNA polymerase II get access to genes. This intensifies mRNA synthesis. On the contrary, deacetylation of histones induces denser DNA packing and chromatin condensation that suppresses transcription.

The levels of acetylation and phosphorylation of histone H3 in the PTS-induced penumbra significantly decreased. Acetylation of lysine 9 in histone H3 stimulates the transcriptional activity of the genome [27]. Oppositely, downregulation of H3K9Ac suppresses protein synthesis. Likewise, acetylation of histone H3 at lysines 9 and 18 occurred in cultured mouse neurons after oxygen and glucose deprivation and following reoxygenation. This caused neuronal apoptosis [12].

The overexpression of histone deacetylases HDAC1, HDAC2 and HDAC4 that was observed in the PTS-induced penumbra caused deacetylation of histone H3. These histone deacetylases are abundant in the brain. In the ischemic brain, HDACs play crucial roles in neurodegeneration. HDAC1 and HDAC2 are localized mainly in the nuclei of neurons, astrocytes and oligodendrocytes, but their expression may change after ischemia [17,28,29]. The outcome of cerebral ischemia depends significantly on the acetylation status of histones. HDAC1 is suggested to be a molecular switch between survival and death in neurons [29]. Its downregulation or inhibition can protect neurons, astrocytes and endothelial cells from hypoxia [30]. Various HDAC inhibitors such as valproic acid, sodium butyrate, suberoylanilide hydroxamic acid or trichostatin A maintain acetylation of histones 3 and 4, attenuate damage and reduce the infarct volume in the ischemic brain. They are currently tested as potential anti-stroke neuroprotectors [13,14,15,31,32,33]. Middle cerebral artery occlusion (MCAO) increased the levels of HDAC1 and HDAC2 in neurons of the ischemic penumbra and in glial cells of white matter in the mouse brain, whereas in the infarct core their levels decreased [28]. However, other authors observed progressive decrease in the HDAC1 and HDAC2 mRNA levels from 3 to 48 h after MCAO, whereas the expression of HDAC3, HDAC6 and HDAC11 substantially increased. This indicated their contribution to stroke pathogenesis [34]. 

In the control cerebral cortex, HDAC1 localized both in the cytoplasm and in the nuclei, whereas HDAC2 was localized exclusively in the nuclei. The similar localization of these HDACs was reported by other authors [17,28,29]. After photothrombotic stroke, HDAC1 appeared in the nuclei of many penumbra cells, where it could contribute into deacetylation of histone H3. HDAC2 was significantly upregulated in the nuclei and appeared in cell processes [35]. Thus, the nuclear localization and overexpression of HDAC2 and HDAC1 could be main reasons of histone H3 deacetylation that could suppress transcription and protein synthesis in the PTS-induced penumbra.

HDAC4 that is highly expressed in neurons usually resides in the cytosol, as in our experiments. In the normal brain it is not bound to chromatin. Its N-terminal domain is required for interaction with the tissue-specific transcription factors and recruitment to some target genes [36,37]. In our experiments, HDAC4 was practically absent in the neuronal nuclei. Its cytoplasmic localization did not change after PTS. One can suggest that it did not contribute to deacetylation of histone H3 in the PTS-induced penumbra. Unlike photothrombotic stroke, MCAO induced the nuclear expression of HDAC4 in the mouse cerebral neurons, reduced acetylation of histones 3 and 4, decreased levels of some downstream pro-survival molecules and neuronal death. However, the cytosolic-restricted HDAC4 did not affect the outcome after ischemia [34]. MCAO was shown to induce re-localization of HDAC4 to the neuronal nuclei in the peri-infarct cortex where it regulated the expression of specific genes in neurons, but not in astrocytes or oligodendrocytes. This effect was associated with the neuronal remodeling but not death, and was observed only from the second day after MCAO. The authors suggested its role in promoting neuronal recovery [37].

The degree of histone acetylation is determined by the balance between the activities of histone deacetylases and histone acetyltransferases [11,38]. When HATs and HDACs work together, genes are maintained in the transcriptionally active state [39]. At 4–24 h after PTS, histone acetyltransferases HAT1 and PCAF were upregulated in the ischemic penumbra along with histone deacetylases. These ubiquitous proteins transfer acetyl group from acetyl-coenzyme A onto histone lysines and play a role as transcription co-activators. Our data showed astrocytic localization of HAT1. In contrast to the opinion that it is the specific nuclear resident [40], we also found HAT1 in the astrocyte cytoplasm and processes (Figure 8). The levels of CBP and histone acetylation were suggested to be required for neuronal resistance against ischemic injury and positive stroke outcome [38].

The observed increase in immunoreactivity of the penumbra tissue to the glial marker GFAP at 4 and 24 h after PTS indicates the early activation of astrocytes after cerebral ischemia. Recent studies have demonstrated the significant role of astrocytes in the brain recovery after injury [41,42]. Brain injury was shown to induce transformation of astrocytes into the “reactive phenotype” with enhanced expression of cytoskeletal proteins and GFAP and changed cell shape.

Thus, the cell fate in the PTS-induced penumbra is determined by the balance between opposite tendencies leading either to histone acetylation and stimulation of protein synthesis, or to histone deacetylation and suppression of protein synthesis. The epigenetic proteins, HATs and HDACs, may be the potential targets for anti-stroke therapy. In fact, a variety of HDAC inhibitors, such as valproic acid, sodium 4-phenylbutyrate, SAHA, trichostatin-A and some others, have shown neuroprotection against ischemic stroke [13,15,31,32,33,43].

In conclusion, photothrombotic stroke, an experimental model of ischemic stroke, reduced acetylation of lysine 9 in histone H3 in the ischemic penumbra at 4–24 h after the impact. This was possibly associated with the increased expression of HDAC1 and HDAC2, but not HDAC4. Deacetylation of histone H3 possibly inhibited protein synthesis in the ischemic penumbra. Simultaneously, histone acetyltransferases HAT1 and PCAF were overexpressed that, oppositely, could facilitate transcription and stimulate protein synthesis.

## 4. Materials and Methods

### 4.1. Chemicals

The Panorama Ab Microarray–Cell Signaling Kits (CSAA1, Sigma-Aldrich Co, St. Louis, MO, United States), antibodies and other chemicals were obtained from Sigma-Aldrich-Rus (Moscow, Russia). Cy3™ or Cy5™ monofunctional reactive dyes were supplied by GE Healthcare.

### 4.2. Animals

The experiments were performed on adult male Wistar rats (200–250 g). The animal holding room was maintained at a temperature of 22–25 °C, 12-h light/dark schedule, and an air exchange rate of 18 changes per hour. All applicable international, national and/or institutional guidelines for the care and use of animals were followed. All experimental procedures were carried out in accordance with the European Union guidelines 86/609/ЕЕС for the use of experimental animals and local legislation for ethics of experiments on animals. The animal protocols were evaluated and approved by the Animal Care and Use Committee of the Southern Federal University (Approval No 02/2014, 12 February 2014).

### 4.3. Focal Photothrombotic Ischemia in the Rat Cerebral Cortex

The unilateral focal photothrombotic infarct (PTI) in the rat somatosensory cortex was induced according to [8,9,10]. The rats were fixed in the stereotactic holder and anaesthetized with chloral hydrate (300 mg/kg, intraperitoneally). The periosteum was removed after the longitudinal incision of the skull skin. Bengal Rose (20 mg/kg) dissolved in sterile physiological saline was injected in the viennae subclavia Then, the somatosensory cortex (0.5 mm anterior to bregma and 3.7 ± 0.5 mm lateral to the midline) was irradiated through the relatively transparent cranial bone by 532 nm diode laser (60 mW/cm^2^, Ø 3 mm, 30 min). The animals were euthanized with the chloral hydrate overdose (600 mg/kg, i.p.) and decapitated by guillotine 1, 4 or 24 h after PTS. The symmetric cortical region in the contralateral hemisphere served as a control. These intervals were chosen because 1 h is the time when the early changes occur and the anti-stroke therapy can start. The 4 h interval corresponds to the “therapeutical window” (2–6 h) when the anti-stroke therapy may be carried out. As shown previously [9,10], the greatest changes in the expression of signaling and neuronal proteins in the penumbra occurred at 4 h after photothrombotic stroke, and decreased (not very significantly) by 24 h.

### 4.4. Proteomic Study

The protein expression profile was studied using “The Panorama Antibody Array–Cell Signaling kit” (CSAA1, Sigma-Aldrich Co, St. Louis, MO, United States) [9]. At 1, 4 or 24 h after PTS, the rat cortex was extracted, and the PTI core was excised using a Ø3 mm circular knife and discarded. Then, the PTS-induced penumbra (a 2-mm cortical ring around the PTI core) was cut out by another Ø7 mm circular knife. The similar piece from the non-irradiated contralateral cortex was used as control. The pieces were weighed, homogenized by ultrasound on ice and lyzed in the Extraction/Labeling buffer supplemented with protease and phosphatase inhibitor cocktails and nuclease benzonase (components of CSAA1). Then, the control and experimental lysates were centrifuged in the cooled centrifuge (rpm, 4 min, 4 °C). The supernatants were frozen in liquid nitrogen and stored at −80 °C for further analysis. After thawing, the protein contents in both experimental and control samples were determined using Bradford reagent. Then, the samples were diluted to 1 mg/mL protein content and incubated 30 min in dark at room temperature with Cy3 or Cy5, respectively. The unbound dye was removed by centrifugation (4000 rpm; 4 min) of the SigmaSpin Columns (CSAA1 components) filled with 200 μL of the labeled protein samples. The eluates were collected, and protein concentration was determined again. In another set, these samples were stained oppositely, by Cy5 and Cy3, respectively.

One microarray was incubated 40 min on a rocking shaker with 5 mL of the mixture of control and experimental samples (10 µg/mL protein each) labeled with Cy3 and Cy5, respectively. Another microarray was incubated with the oppositely labeled samples: Cy5 and Cy3, respectively. Such swapped staining provided verification of results and compensation of a potential bias in binding of Cy3 or Cy5 dyes to protein samples. This ensures the double test and full control of the experiment [44]. After following triple washing in the washing buffer (CSAA1 component) and triple washing in pure water, the microarray slides were air dried overnight in dark.

These microarrays were scanned on the Molecular Devices GenePix Microarray Scanner 4100A (Molecular Devices, Sunnyvale, CA, USA) at 532 and 635 nm (fluorescence maximums of Cy3 and Cy5, respectively). The integrated fluorescence intensity in each antibody spot was proportional to the quantity of the bound protein. The fluorescence images of the antibody microarrays were normalized (ratio-based normalization) and analyzed using the software GenePix Pro 6.0. The fluorescence of small rings around each spot was subtracted as a background. The median fluorescence intensity over all spot pixels represented the protein content in each spot. The ratios of the experimental to control values characterized the difference in the level of the appropriated protein between photothrombotic and control cortical tissue in the contralateral cortex. Two samples, labeled independently and reversely in duplicate, provided four experimental values for each protein. The experiments were repeated four times and the experiment/control ratios were averaged. *n* = 16 (4 animals × 4 values in each experiment). The Student’s t-test and 95% significance level were used. Only proteins whose expression differed more than 30% from the control (the cutoff level) were used. The data on Figure 1, Figure 2 and Figure 3 are presented as means ± SD.

### 4.5. Immunofluorescence Microscopy

At 1, 4 or 24 h after PTS rats were anaesthetized by chloral hydrate and transcardially perfused with 10% formalin. The brains were extracted, post-fixed in formalin overnight and then placed into 20% sucrose in PBS for 48 h at 4°С. Brain frontal sections (20 μm thick; + 4.7 mm from the bregma to −2.1 mm [44] were obtained using the Leica VT 1000 S vibratome (Germany). They were frozen in 2-methylbutane and stored at −80 °С. After thawing, the slices were washed in PBS. Nonspecific binding of antibodies was blocked by 5% BSA with 0.3% Triton X-100 in PBS (1 h, room temperature). Then, the slices were incubated overnight at 4°С in the same BSA solution with the rabbit antibodies (all from Sigma-Aldrich): anti-Histone H3 Acetyl-Lys9 (anti-H3K9Ac; SAB4500347; 1:250), anti-HDAC1 (H3284; 1:500), anti-HDAC2 (H3159; 1:500), anti-HDAC4 (SAB4300413; 1:250), anti-HAT1 (SAB4503405; 1:250), anti-PCAF (1:250) or mouse antibody anti-GFAP (SAB4200571; 1:1000). After triple 5-min washing in PBS, the slices were incubated for 1 h with the fluorescence-labeled secondary antibody anti-Rabbit CF488A (SAB4600045; 1:1000). Then, the slices were mounted in 60% glycerol. Negative control: omission of primary antibodies. The slices were analyzed using a microscope Eclipse FN1 (Nikon, Japan). Green protein fluorescence was registered using the excitation wavelength 450–490 nm and the long-path filter 505 nm. Red marker fluorescence was registered using excitation wavelength 510–560 nm and the long-path filter 575 nm.

Quantitative estimation of the fluorescence of experimental and control preparations was carried out using 10–15 images obtained with the same camera settings. The average fluorescence intensity in the area occupied by the cells was determined in each image using the Image J software. The corrected total cell fluorescence intensity (CTCF) that was proportional to levels of the protein expression was calculated as follows: *I = I_id_ − A_c_*I_b_*; where *I*—the corrected intensity of total cell fluorescence (CTCF), *I_id_*—integrated fluorescence intensity, *A_c_*—area of selected cell, *I_b_*—mean background fluorescence. The relative changes *ΔI* of the corrected cell fluorescence in penumbra comparing with that in control cortex was calculated as: *ΔI = (I_pen_ − I_c_)/I_c_*, where *I_pen_* is the fluorescence intensity in the penumbra and *I_c_* is the fluorescence intensity in control samples. Immunoreactivity of GFAP was quantified by measuring the integrated optical density (ID—intensity of fluorescence per unit of surface area). The co-localization of HDAC2 and HAT1 with the astrocyte marker (GFAP) was evaluated using ImageJ (http://rsb.info.nih.gov/ij/) with the JACoP plugin [45]. The Manders’ co-localization coefficient M1reflects the fraction of pixels containing red (GFAP) and green (proteins) signals in relation to the overall signal registered in the red channel [46]. No less than three fields of visions have been analyzed for each brain region. The intergroup comparisons were carried out using one-way ANOVA.

## Figures and Tables

**Figure 1 ijms-20-02882-f001:**
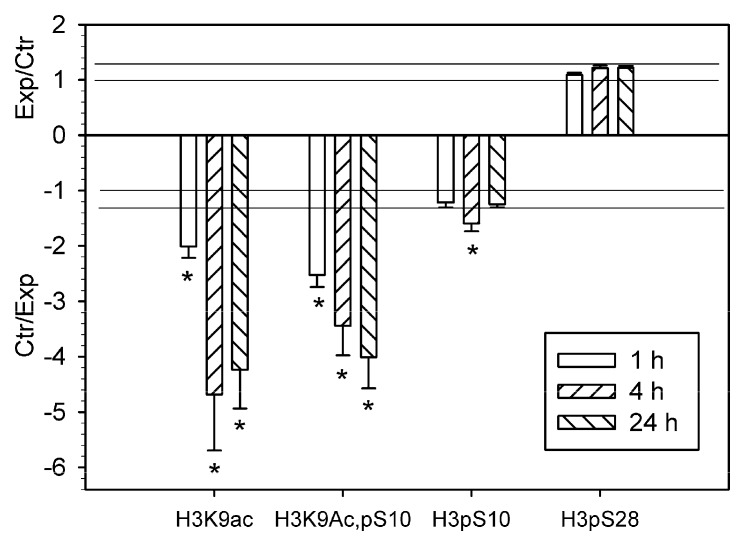
The ratios of the levels of acetylated or phosphorylated histone H3 in the ischemic penumbra 1, 4 or 24 h after photothrombotic stroke in the rat cerebral cortex (Exp) to that in the untreated contralateral cortex of the same animals (Ctr). H3K9Ac—histone H3 acetylated on lysine 9; H3K9Ac,pS10—histone H3 acetylated on lysine 9 and phosphorylated on serine 10; H3pS10—histone H3 phosphorylated on serine 10; H3pS28—histone H3 phosphorylated on serine 28. The antibody microarray data. *n* = 16 (4 animals × 4 values in each experiment). Mean ± SD. * *p* < 0.05.

**Figure 2 ijms-20-02882-f002:**
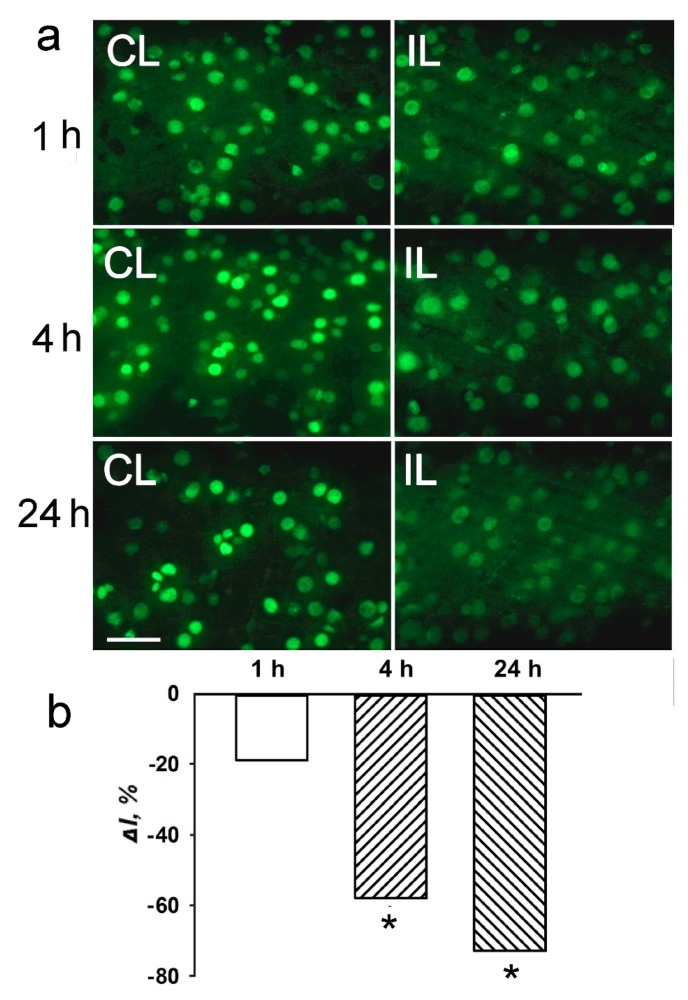
The cellular localization and expression of acetylated histone H3 in the penumbra at 1, 4 or 24 h after photothrombotic stroke in the rat cerebral cortex. (**a**) The typical immunofluorescence images of H3K9Ac in the ischemic penumbra (IL, right column) and control contralateral cortex (CL, left column). Scale bar, 20 μm. (**b**) Percent changes in fluorescence intensity of H3K9Ac-positive cells in the ischemic penumbra 1, 4 or 24 h after photothrombotic stroke in the rat cerebral cortex relatively to that in the control contralateral cortex. *ΔI %* is the mean corrected total cell fluorescence (CTCF) averaged over penumbra minus mean control CTCF/mean control CTCF × 100% (experiment versus control). *n* = 5. * *p* < 0.05.

**Figure 3 ijms-20-02882-f003:**
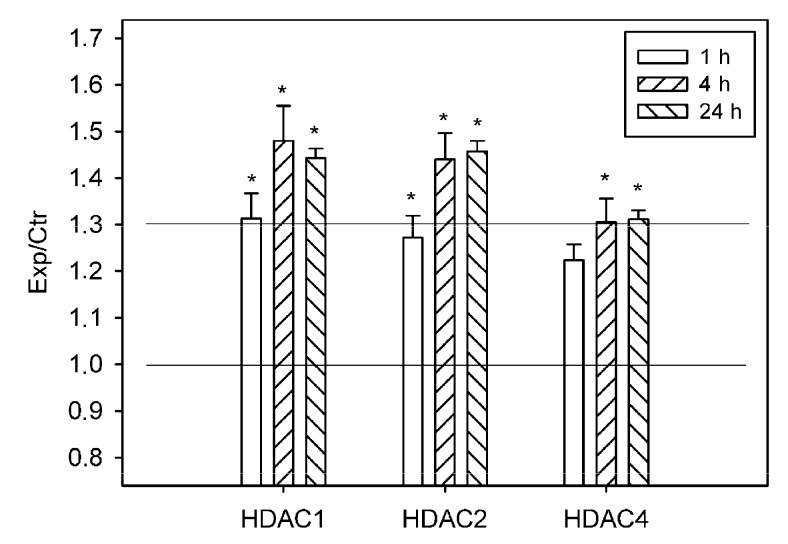
The ratios of levels of histone deacetylases HDAC1, HDAC2, and HDAC4 in the ischemic penumbra 1, 4, or 24 h after photothrombotic stroke in the rat cerebral cortex (Exp) to that in the untreated contralateral cortex of the same animals (Ctr). The antibody microarray data. *n* = 16 (4 animals × 4 values in each experiment). Mean ± SD. * *p* < 0.05.

**Figure 4 ijms-20-02882-f004:**
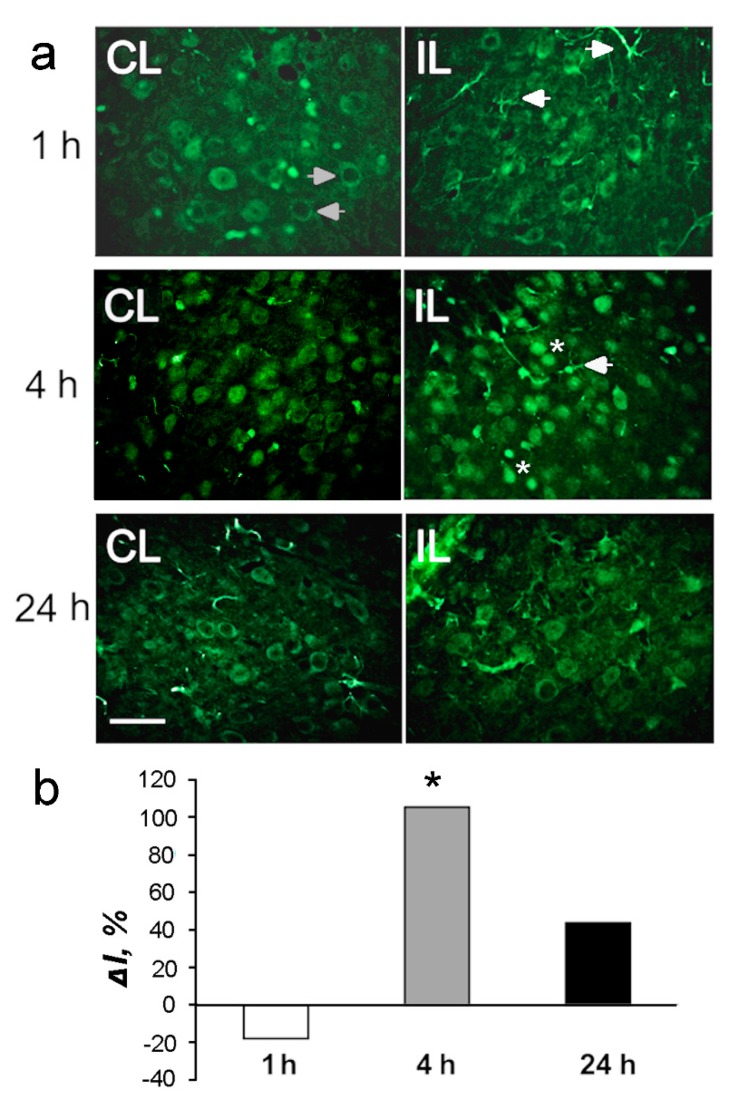
The cellular distribution and expression of histone deacetylase HDAC1 in the penumbra at 1, 4 or 24 h after photothrombotic stroke in the rat cerebral cortex. (**a**) The typical immunofluorescence images of HDAC1 localization in the ischemic penumbra (IL, right column) and in the control contralateral cortex (CL, left column). Arrows in the left column indicate cells with cytoplasmic HDAC1, arrows in the right columns show the expression of HDAC1 in cellular processes. Asterisks—neuronal nuclei. Scale bar, 20 μm. *(***b***)* Percent changes in the fluorescence intensity of HDAC1-positive cells in the ischemic penumbra 1, 4 or 24 h after photothrombotic stroke (PTS) in the rat cerebral cortex relatively to that in the contralateral cortex. *ΔI %* is the mean corrected total cell fluorescence (CTCF) averaged over penumbra minus mean control CTCF/mean control CTCF × 100% (experiment versus control). *n* = 5. * *p* < 0.05.

**Figure 5 ijms-20-02882-f005:**
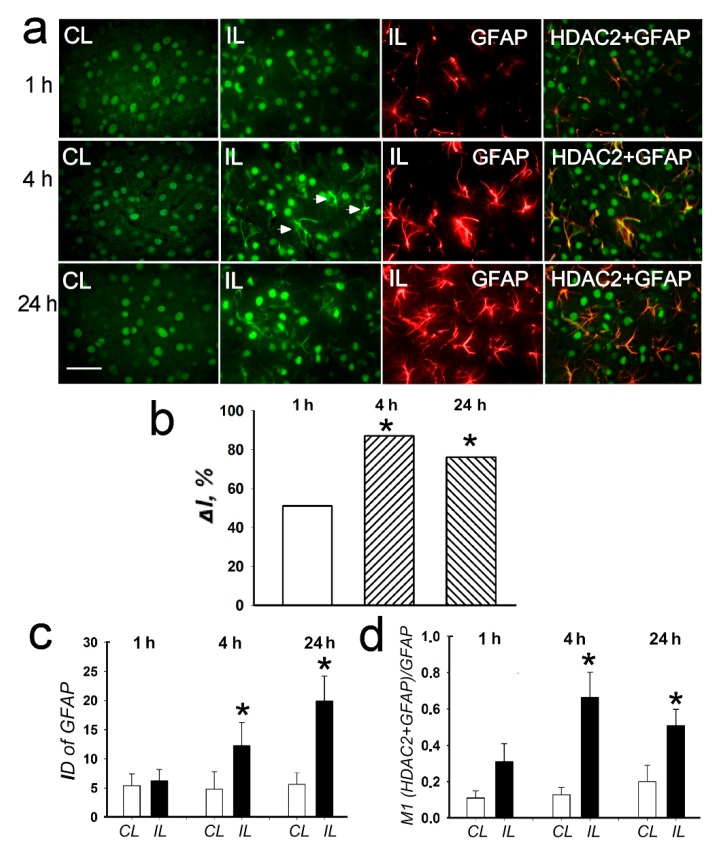
The cellular distribution of histone deacetylase HDAC2 in the penumbra at 1, 4 or 24 h after PTS in the rat cerebral cortex. (**a**) The typical immunofluorescence images of HDAC2 localization (green) in the ischemic penumbra (IL) and in the control contralateral cortex (CL), the expression of astroglia marker GFAP (red) and merged images of HDAC2 and GFAP. HDAC2 fluorescence in GFAP-labeled astrocytes is yellow. White arrows—HDAC2 expression in astrocytes. Scale bar, 20 μm. (**b**) Percent changes in fluorescence intensity of HDAC2-positive cells in the ischemic penumbra 1, 4 or 24 h after photothrombotic stroke in the rat cerebral cortex relatively to that in the control contralateral cortex. *ΔI %* is the mean corrected total cell fluorescence (CTCF) averaged over penumbra minus mean control CTCF/ mean control CTCF × 100% (experiment versus control). (**c**) Integrated optical density of GFAP-positive cells (ID—fluorescence intensity per a unit of surface area). (**d**) Manders’ coefficients M1 display PTS-induced changes in co-localization of HDAC2 with GFAP. *n* = 7. * *p* < 0.05.

**Figure 6 ijms-20-02882-f006:**
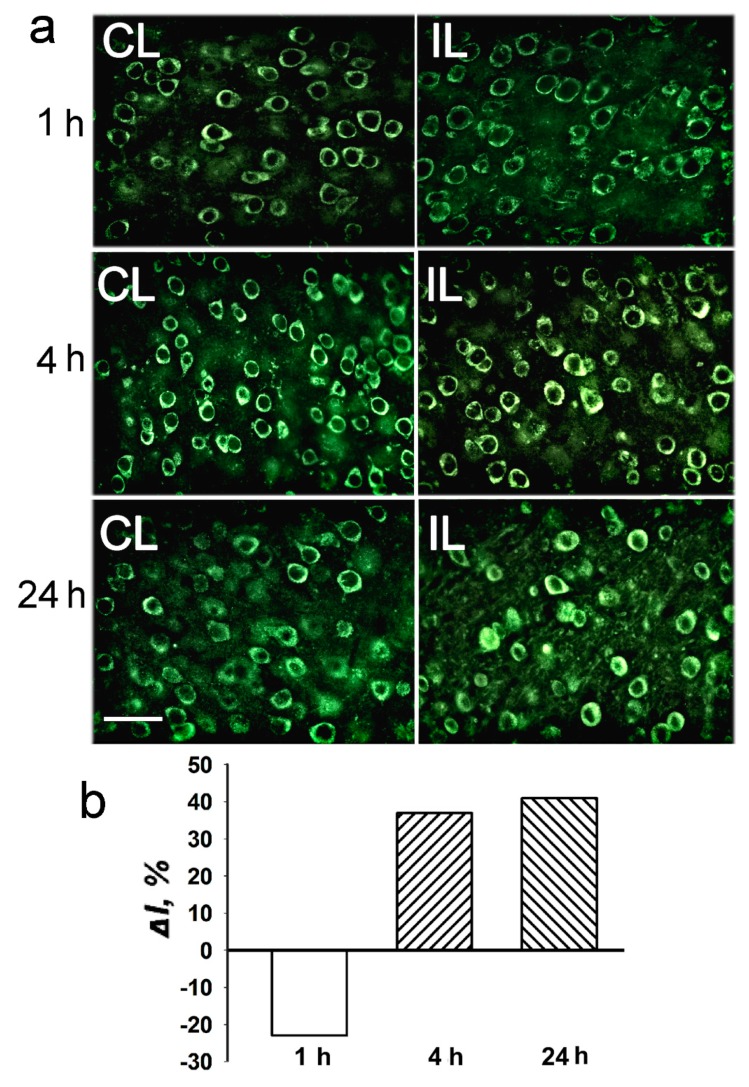
The cellular distribution of histone deacetylase HDAC4 in the penumbra at 1, 4 or 24 h after photothrombotic stroke in the rat cerebral cortex. (**a**) The typical immunofluorescence images of the HDAC4 localization in the ischemic penumbra (IL, right column) and control contralateral cortex (CL, left column). Scale bar, 20 μm. (**b**) Percent changes in fluorescence intensity of HDAC4-positive cells in the ischemic penumbra 1, 4 or 24 h after photothrombotic stroke in the rat cerebral cortex relatively to that in the control contralateral cortex. *ΔI %* is the mean corrected total cell fluorescence (CTCF) averaged in the penumbra minus mean control CTCF/mean control CTCF × 100% (experiment versus control). *n* = 5.

**Figure 7 ijms-20-02882-f007:**
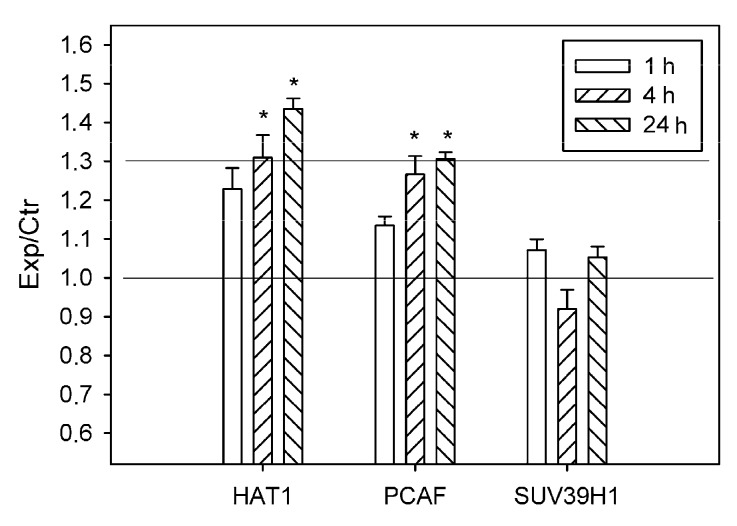
The ratios of levels of histone acetyltransferases HAT1 and PSAF, and histone-lysine N-methyltransferase SUV39H1 in the ischemic penumbra 1, 4 or 24 h after photothrombotic stroke in the rat cerebral cortex (Exp) to those in the untreated contralateral cortex of the same animals (Ctr). The antibody microarray data. *n* = 16 (4 animals × 4 values in each experiment). Mean ± SD. * *p* < 0.05.

**Figure 8 ijms-20-02882-f008:**
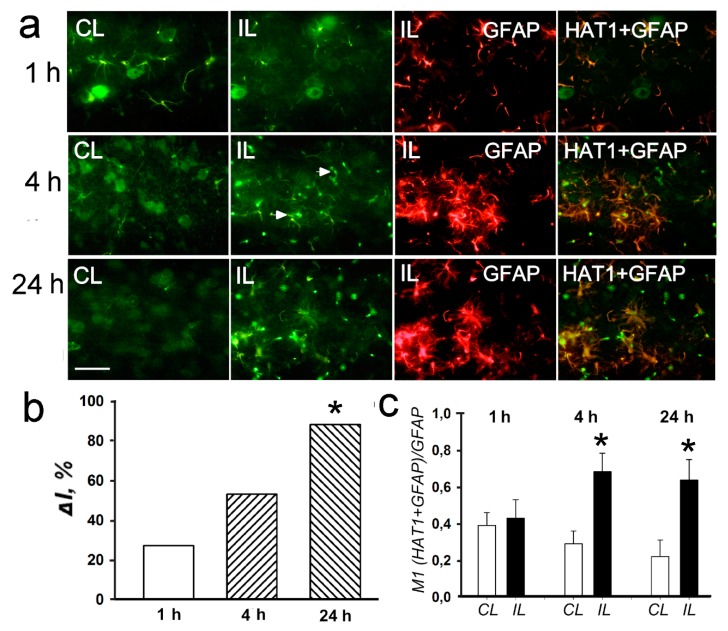
The cellular distribution and expression of histone deacetylase HAT1 in the penumbra at 1, 4 or 24 h after PTS in the rat cerebral cortex. (**a**) The typical immunofluorescence images of HAT1 expression (green) in the ischemic penumbra (IL) and in the control contralateral cortex (CL), the expression of astroglia marker GFAP (red) and merged images of HAT1 and GFAP. HDAC2 fluorescence in GFAP-labeled astrocytes is yellow. White arrows—HAT1 expression in astrocytes. Scale bar, 20 μm. (**b**) Percent changes in fluorescence intensity of HAT1-positive cells in the ischemic penumbra 1, 4, or 24 h after photothrombotic stroke in the rat cerebral cortex relatively to that in the control contralateral cortex. *ΔI %* is mean corrected total cell fluorescence (CTCF) averaged in the penumbra minus mean control CTCF/mean control CTCF × 100% (experiment versus control). (**c**) Manders’ coefficients M1 that display the PTS-induced changes in co-localization of HAT1 with GFAP. *n* = 7. * *p* < 0.05.

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
