# Peer review of "Epigenetic Alterations Induced by Photothrombotic Stroke in the Rat Cerebral Cortex: Deacetylation of Histone h3, Upregulation of Histone Deacetylases and Histone Acetyltransferases"

_ijms, 2019, doi:10.3390/ijms20122882_

Reviewer 1 Report

I have 3 comments.

1) Some works have been focused on the epigenetic processes within the ischemic zone. But lacking of citation2) title might be misleading, a change of title is needed to reflect the result.

3) 1hr, 4hr, 24hr were chosen to determine such proteins. Why these 3 intervals were chosen? there is a large time interval between 4hr and 24 hr. Since it is "epigenetics" proteins, which the level/expression can change drastically, how are you be so sure that this is the best tine interval? perhaps 1hr, 4hr, 8hr, 24 hr, might be a better choice. please justify.

Author Response

Thank you for important comments

 I have 3 comments.

1) Some works have been focused on the epigenetic processes within the ischemic zone. But lacking of citation

We cite works devoted to the role of epigenetic processes in cerebral ischemia and to protective effects of HDAC inhibitors:

References 11-13, 22,23,25,26, 28, 31,32,34,35

2) title might be misleading, a change of title is needed to reflect the result.

We corrected the title as follows:

Epigenetic alterations induced by photothrombotic stroke in the rat cerebral cortex: deacetylation of histone H3, upregulation of histone deacetylases and histone acetyltransferases”

3) 1hr, 4hr, 24hr were chosen to determine such proteins. Why these 3 intervals were chosen? there is a large time interval between 4hr and 24 hr. Since it is "epigenetics" proteins, which the level/expression can change drastically, how are you be so sure that this is the best tine interval? perhaps 1hr, 4hr, 8hr, 24 hr, might be a better choice. please justify.

The following text is added to Material and Methods, the Focal photothrombotic ischemia in the rat cerebral cortex section:

These intervals were chosen because 1h is the time when the early changes occur and the anti-stroke therapy can start. The 4h interval corresponds to the “therapeutical window” (2-6 h) when the anti-stroke therapy may be carried out. As shown previously [9-10], the greatest changes in the expression of signaling and neuronal proteins in the penumbra occurred at 4h after photothrombotic stroke, and decreased (not very significantly) by 24 h.

As shown in the present work, the stronger epigenetic changes occurred at 4h post-stroke, and did not change significantly at 24 h. So, we did not anticipate any significant difference at 8h interval.

Reviewer 2 Report

The presented manuscript describes the changes of different epigenetic processes in ischemic penumbra induces by photothrombotic stroke (PTS) during first 24 hours after the PTS treatment. The action involved in the rat cerebral cortex after administration of Rose Bengal, induced local vessel occlusion followed by epigenetic changes, chromatin decondensation and decreasing in its transcriptional activity. The topic addressed by the Authors sounds quite interesting and carries the aspects of scientific novelty in view of new data on photothrombotic stroke studies. I my opinion the article is very good wrote with the detailed designing of experiments and good results discussion. It deserves to publish in International Journal of Molecular Sciences without any changes.

Author Response

Thank you for important comments

Reviewer 3 Report

This study investigates in a photothrombotic stroke (PTS) model of rats whether the activity of histone deacetylase and acetyltransferase and thus the degree of histone acetylation changes in the perilesional area of the PTS in the acute phase (i.e., at three time points within 24 hours after stroke induction). Over all, they found that the acetylation and phosphorylation of histone H3 in the perilesional area was reduced, whereas histone deacetylases HDAC1, HDAC2 and, to lesser extent, HDAC4, as well as histone acetyltransferases HAT1 were upregulated.

There are some issues that authors should address:

There are some orthographical and syntax errors throughout the manuscript.

The authors investigate one of the five histones, i.e., histone 3. What was the reason to investigate especially histone 3 and not one of the other types? Is this histone most relevant in brain injury such as stroke? The authors are encouraged to outline in the introduction why they focused on this histone (especially in the context of stroke).

 “… contralateral cortex was used as control”. (p 12, line 338). Stroke may also affect remote areas in the brain. Thus, I wonder whether samples of the contralateral hemisphere are an optimal control. Why did the authors not use sham-operated animals as controls?

How many animals did the authors use in this study? Please clarify.

The authors should indicate how many samples (or animals) they have used per analysis.

The photothrombotic stroke model results only in a small penumbra. Therefore, the term “perilesional area” is probably more correct than penumbra. The small (or even lacking) penumbra in the PTS-model might also be described as a study limitation in the end of the discussion.

What is the meaning of asterisks in figure 4 (right column, middle row)? Please clarify this issue in the figure legend.

The presented immunofluorescence images are not really convincing. The cells demonstrated in the panels have the shape of immune cells (or the green dots are only nuclei, e.g., figure 4, right row, middle panel or figure 5). Immune cells are often accumulated in the perilesional area. Thus, how can we be sure that – in this context - cerebral cells demonstrate a different expression or activity of HDAC and not immune cells? Therefore, double staining with HDAC and NeuN (neuronal marker) and DAPI or Hoechst (marker for nuclei) is highly recommended.

Author Response

Thank you for important comments

There are some issues that authors should address:

There are some orthographical and syntax errors throughout the manuscript.

Thank you, we tried to correct the errors

The authors investigate one of the five histones, i.e., histone 3. What was the reason to investigate especially histone 3 and not one of the other types? Is this histone most relevant in brain injury such as stroke? The authors are encouraged to outline in the introduction why they focused on this histone (especially in the context of stroke).

We added the following paragraph in the Introduction:

Different epigenetic processes such as DNA methylation, acetylation, methylation, phosphorylation and other covalent modification of histones can regulate gene expression and production of diverse proteins in the cell (Allis et al., 2006). Histone acetylation that causes chromatin decondensation facilitates transcriptional processes (Kouzarides and Berger, 2006). This is a master-regulator of gene expression. Proteins involved in these epigenetic processes may be promising targets for development of anti-stroke agents. Cerebral ischemia was shown to reduce acetylation of histones H3 and H4 and thereby to suppress protein biosynthesis. Some chemical agents recovered the optimal histone H3 acetylation and were tested as drugs for stroke therapy (Lansilotta, 2013; Hu et al., 2016; Schweizer , 2013; Zhao 2016; Jhelum et al., 2017). In the present work we concentrated on acetylation and phosphorylation of histone H3. Acetylation of lysine 9 in histone H3 (H3K9ac) was of particular interest because of its location mainly in the promoter regions near the sites of gene transcription start (Wang et al., 2008).

“… contralateral cortex was used as control”. (p 12, line 338). Stroke may also affect remote areas in the brain. Thus, I wonder whether samples of the contralateral hemisphere are an optimal control. Why did the authors not use sham-operated animals as controls?

We used both sham-operated and contralateral cortex controls in our previous work (Demyanenko et al., 2018), but did not observe any difference between them in studies of the expression of HDAC1, HDAC2, and HDAC8 in the mouse cerebral cortex and hippocampus at 3 but not 7 days after photothrombotic stroke. Therefore, we suggested the absence of any difference in shorter periods and used only contralateral control.

How many animals did the authors use in this study? Please clarify.

The authors should indicate how many samples (or animals) they have used per analysis.

Corrected in the figure legends:

In proteomic experiments (figures 1, 3 and 7) n=16 (4 animals * 4 values in each experiment).

For fig.2,4, and 6: n = 5.

For fig.5 and 8: n = 7.

The photothrombotic stroke model results only in a small penumbra. Therefore, the term “perilesional area” is probably more correct than penumbra. The small (or even lacking) penumbra in the PTS-model might also be described as a study limitation in the end of the discussion.

In the previous morphological and ultrastructural study (Uzdensky et al., 2017) we showed that reduced laser light intensity, but prolonged exposure provide the relatively wide (1.5 mm width) peri-infarct zone. So, we used this traditional term – penumbra

We added to Discussion the following paragraph:

Various models of ischemic stroke such as ligations of main cerebral arteries, or bloodstream occlusion by the inserted nylon thread, natural or artificial emboli are currently studied. All of them display various sides of the cerebral ischemic process. PTS as a model of ischemic stroke has some advantages such as well-defined injury size and location, low invasiveness, low animal mortality, and high reproducibility. Small penumbra width was considered by some researchers as a drawback of this model [17]. However, our previous morphological and ultrastructural study (Uzdensky et al., 2017) showed that less intense, but prolonged laser light exposure provided relatively wide (1.5 mm width) penumbra produced by photothrombotic stroke in the rat brain cortex.

What is the meaning of asterisks in figure 4 (right column, middle row)? Please clarify this issue in the figure legend.

Indicated:

Asterisks – neuronal nuclei.

The presented immunofluorescence images are not really convincing. The cells demonstrated in the panels have the shape of immune cells (or the green dots are only nuclei, e.g., figure 4, right row, middle panel or figure 5). Immune cells are often accumulated in the perilesional area. Thus, how can we be sure that – in this context - cerebral cells demonstrate a different expression or activity of HDAC and not immune cells?.

Neuronal cells and their nuclei are well seen in these preparations, especially when both cytoplasm and nuclei are displayed as in fig.6.  Neurons occupy the major part of the images. Neuronal nuclei are bigger than the nuclei of astrocytes (fig.5,8). They look as circles, not dots (fig.2, 5, 6 and asterisks in fig.4).

Therefore, double staining with HDAC and NeuN (neuronal marker) and DAPI or Hoechst (marker for nuclei) is highly recommended

We used the recommended staining methods in the recent work (Demyanenko et al., 2018) and in current studies (unpublished). These data confirm the present conclusions.

Reviewer 4 Report

In this manuscript, authors showed that photothrombotic stroke causes the upregulation of histone deacetylases HDAC1 and HDAC2, which is associated with reduced acetylation of histone H3.

After careful reading the reviewer has made following comments:

1. Why western blot in not preferred for proteins level determination?

2. Different colors for figures 3 and 7 will improve the figures

3. Abstract says there is no upregulation of HDAC-4, but lines 80 and 102 mention the upregulation of HADAC4.

4. Did authors observe any symptoms in mice after photothrombotic stroke? As there is upregulation of HDAC1 and HDAC2, did they treat the mice with HDAC1 and/or HDAC2 inhibitors?

5. If which form the rose Bengal was injected? Formulation or solution in buffer/DMSO/Water?

6. Wavelength range or region should be mentioned for all the microscopy experiments.

The manuscript will be suitable for publication after minor revisions.

Author Response

Thank you for important comments

After careful reading the reviewer has made following comments:

1. Why western blot in not preferred for proteins level determination?

Protein levels were determined in the proteomic study. It was also estimated by fluorescence microscopy, which also shows the expression of proteins in different cell types and allows determining of penumbra borders

2. Different colors for figures 3 and 7 will improve the figures

We used hatching to display the columns for black-and-white monochrome print.

3. Abstract says there is no upregulation of HDAC-4, but lines 80 and 102 mention the upregulation of HADAC4.

Yes, we added in the text:

According to the proteomic (Figure 3) and immunofluorescence microscopy data (Figure 6), its level in the PTS-induced penumbra increased by 30-40% (p<0.05 and p>0.05, respectively) less than that for HDAC 1 and HDAC2.

Added in the Abstract:

This was associated with upregulation of histone deacetylases HDAC1 and HDAC2, but not HDAC4, which did not localize in the nuclei.

4. Did authors observe any symptoms in mice after photothrombotic stroke? As there is upregulation of HDAC1 and HDAC2, did they treat the mice with HDAC1 and/or HDAC2 inhibitors?

Yes we use two behavior tests and study the effects of some HDAC inhibitors on mice. These studies are currently underway and their results are being prepared for publication.

5. If which form the rose Bengal was injected? Formulation or solution in buffer/DMSO/Water?

Corrected in Methods:

Bengal Rose (20 mg/kg) dissolved in sterile physiological saline was injected in the v. subclavia.aline

6. Wavelength range or region should be mentioned for all the microscopy experiments.

Inserted in Methods:

Green protein fluorescence was registered using the excitation wavelength 450-490 nm and the long-path filter 505 nm. Red marker fluorescence was registered using excitation wavelength 510-560 nm and the long-path filter 575 nm.